# Bacteriocin Production by *Escherichia coli* during Biofilm Development

**DOI:** 10.3390/foods11172652

**Published:** 2022-09-01

**Authors:** Hanna Fokt, Sara Cleto, Hugo Oliveira, Daniela Araújo, Joana Castro, Nuno Cerca, Maria João Vieira, Carina Almeida

**Affiliations:** 1Centre of Biological Engineering (CEB), Campus de Gualtar, University of Minho, 4710-057 Braga, Portugal; 2LABBELS–Associate Laboratory, 4710-057 Braga, Portugal; 3INIAV, IP-National Institute for Agrarian and Veterinary Research, Rua dos Lagidos, Lugar da Madalena, 4485-655 Vila do Conde, Portugal; 4LEPABE-Laboratory for Process Engineering, Environment, Biotechnology and Energy, Faculty of Engineering, University of Porto, Rua Dr. Roberto Frias, 4200-465 Porto, Portugal

**Keywords:** *E. coli*, biofilm, bacteriocin, colicin, antibacterial, small molecules

## Abstract

*Escherichia coli* is a highly versatile bacterium ranging from commensal to intestinal pathogen, and is an important foodborne pathogen. *E. coli* species are able to prosper in multispecies biofilms and secrete bacteriocins that are only toxic to species/strains closely related to the producer strain. In this study, 20 distinct *E. coli* strains were characterized for several properties that confer competitive advantages against closer microorganisms by assessing the biofilm-forming capacity, the production of antimicrobial molecules, and the production of siderophores. Furthermore, primer sets for *E. coli* bacteriocins–colicins were designed and genes were amplified, allowing us to observe that colicins were widely distributed among the pathogenic *E. coli* strains. Their production in the planktonic phase or single-species biofilms was uncommon. Only two *E. coli* strains out of nine biofilm-forming were able to inhibit the growth of other *E. coli* strains. There is evidence of larger amounts of colicin being produced in the late stages of *E. coli* biofilm growth. The decrease in bacterial biomass after 12 h of incubation indicates active type I colicin production, whose release normally requires *E. coli* cell lysis. Almost all *E. coli* strains were siderophore-producing, which may be related to the resistance to colicin as these two molecules may use the same transporter system. Moreover, *E. coli* CECT 504 was able to coexist with *Salmonella enterica* in dual-species biofilms, but *Shigella dysenteriae* was selectively excluded, correlating with high expression levels of colicin (E, B, and M) genes observed by real-time PCR.

## 1. Introduction

Food contamination by pathogenic *Escherichia coli* has been a serious public health problem and has caused huge economic losses worldwide [1,2,3]. Despite the fact that *E. coli* inhabits normal microbiota in the gut, and in most cases is considered harmless, certain pathogenic strains of *E. coli* can infect the gut and cause severe illness. Typically, pathogenic *E. coli* infection causes severe diarrhea. Furthermore*, E. coli* is also the most common enteric organism causing extraintestinal infections in humans, particularly the urinary tract, peritoneum, and blood infections [4,5,6,7]. It is also important to highlight that *E. coli* species are able to prosper in single and multispecies biofilm environments [8,9,10].

In nature, biofilms are the most common lifestyle of bacteria, consisting of complex sessile communities adhered to a variety of biotic and abiotic surfaces [11]. Biofilms are characterized by the presence of a matrix, which enfolds bacterial cells and contains extracellular polysaccharides, proteins, and extracellular DNA (eDNA) [12]. These features provide bacteria protection and stability in a changing environment, making them significantly more resistant to antimicrobials than planktonic cells [13,14]. It may also facilitate bacterial survival in the host, thus potentially being related to pathogenicity [15,16,17].

The ability of bacteria to secrete bacteriocins for partial or complete inhibition of their counterparts’ growth can be regarded as another competitive advantage [18,19,20]. Furthermore, the production of antimicrobial peptides seems to be a common phenomenon among bacteria [21,22]. Additionally, bacterial antimicrobials usually have a narrow target spectrum and are only active against species closely related to the producer [23,24].

The most widely studied bacteriocins are colicins, produced by *E. coli*. These large, plasmid-encoded proteins, of two main types, I and II, are induced by the bacterial SOS system and normally involve producer-cell suicide [25,26]. Their mechanism of action is still unclear, but it is known that some types of colicins may interact directly with siderophores’ cell membrane receptors [27,28,29]. Siderophores are iron-chelating molecules that have a key role in sequestering iron from the surrounding environment, a capacity that is essential for bacteria to survive in low ferrous iron conditions [30]. Microcins, another family of bacteriocins, are also produced by *E. coli* [21,29,31], are small chromosome-encoded peptides, are not regulated by SOS regulon, and do not lead to producer-cell death [26].

There are many studies concerning the optimization of bacteriocin production in order to understand how it is affected by different physical and chemical factors, both in biofilm and in planktonic phase growth [24,32,33,34,35]; however, to our best knowledge, there is a lack of studies characterizing bacteriocin production with biofilm development. The present study aimed to investigate the biofilm formation capacity of *E. coli* strains and relate this with their ability to produce bacteriocins and to outcompete related species.

## 2. Materials and Methods

### 2.1. Bacterial Strains and Culture Conditions

Bacterial strains used in this study are listed in Appendix A. *E. coli* K12, *E. coli* K12 Δ*impA* mutant [36] with increased permeability due to incomplete outer membrane, and *E. coli* ER1100A Δ*entF* defective in siderophore production [37] were used as sensitive strains in antimicrobial activity assay. For planktonic cultures, 200 mL of bacterial culture were grown at 37 °C, 120 rpm in a complete Luria-Bertani (LB) medium (Liofilchem, Roseto degli Abruzzi, Italy) for up to 72 h [38]. All strains were maintained in LB agar medium at 4 °C.

### 2.2. Biofilm Formation Assay

For all biofilm experiments, cells were pre-grown in LB medium overnight (for approximately 18 h), at 37 °C, 120 rpm [39]. Overnight cultures were diluted in order to obtain a final concentration of approximately 10^7^ CFU/mL. LB in the presence and absence of 0.25% (*w*/*v*) glucose (Liofilchem, Roseto degli Abruzzi, Italy) was used to form the biofilms to evaluate if glucose would potentate the biofilm formation. Untreated polystyrene 24- and 96-well cell-tissue microtiter plates (Orange Scientific, Braine-l’Alleud, Belgium) were filled, respectively, with 2 mL (4 replicates per strain) and 200 µL (8 replicates per strain) of bacterial suspension per well. For dual-species biofilms, an additional 1:2 dilution was performed. Volumes of 1 mL (24-well plate) or 100 µL (96-well microtiter plate) of the different bacterial species suspension were mixed in order to obtain a final concentration of 10^7^ CFU/mL. The plates were sealed with an air-permeable cover sheet (VWR, Radnor, PA, USA) and incubated up to 72 h at 37 °C, 120 rpm. Medium replacement was performed every 24 h. Culture liquid with planktonic cells was removed to leave a layer of adhered cells and the same volume of fresh medium was added to each well. At different sampling times (6, 12, 24, 48, and 72 h), the spent medium was carefully removed. Adherent cells were washed, and biofilm formation was assessed by the crystal violet assay, as described previously by O’Toole and Kolter, 1998 [40]. To estimate total biofilm biomass, the optic density (OD) was measured at 570 nm [39]. This experiment was performed in triplicate for each of the species used.

### 2.3. Antimicrobial Activity Assay

For the bacteriocin quantification, the 24 h old biofilm was considered the time point 0. This assured us that only bacteriocin from biofilm, and not from the planktonic phase, was quantified. Every 24 h, the biofilm was washed as described above and the medium was replaced. The antimicrobial activity was measured for 6, 12, 24, 48, and 72 h. The crude extract or also called the cell-free supernatant (CFS) was obtained by centrifugation of each of the 8 mL samples taken at different sampling times. The centrifugation was performed at 6500*× g*, 4 °C for 10 min. Following this, the supernatants were filter-sterilized (0.2 µm acrodisc, Orange Scientific, Braine-l’Alleud, Belgium). To obtain the supernatant samples from planktonic cultures, the same procedure was performed.

Antimicrobial activity of cell-free spent media was assessed on lawns of different strains of *E. coli* (listed in Appendix A), *Staphylococcus aureus* CECT 86, *Staphylococcus epidermidis* CECT 4184, *Salmonella enterica* serovar Enteritidis SGSC 2476, *Shigella dysenteriae* ATCC 11335, and *Listeria monocytogenes* CECT 5873 using the soft agar technique [41] as follows: sterile soft agar (0.5% (*w*/*v*) agar) was melted to 45 °C and mixed with 50 µL (approximately 5 × 10^5^ CFU/mL) of the appropriate bacterial suspension (bacteria grown overnight as previously described). Then, 2.5 mL of the referred agar-bacteria suspension were poured over Petri dishes containing LB agar medium and allowed to solidify. Filter-sterilized spent media from biofilms and planktonic cultures were subjected to serial dilutions (1/2, 1/14, 1/8, 1/16, 1/32, 1/64, 1/128, 1/256, 1/512), and 5 µL were spotted onto the agar and allowed to air-dry. The plates were incubated overnight at 37 °C, and at the end of the incubation period, the microbial growth inhibition zone was analyzed. These experiments were performed in triplicate.

### 2.4. Culturability Assessment of Dual-Species Biofilms

Dual-species biofilms were grown up to 72 h as described above, washed two times with 0.9% (*w*/*v*) NaCl (Merck, Darmstadt, Germany), and sonicated for 30 min for release of attached cells. To assess the different populations, samples were then plated with the appropriate dilutions in MacConkey agar (Liofilchem, Roseto degli Abruzzi, Italy) plates in triplicate. This medium allows discrimination between *S. enterica*/*S. dysenteriae* and *E. coli* based on each species’ ability to degrade lactose. The MacConkey plates were incubated at 37 °C overnight, and colony-forming units (CFUs/mL) were counted as described by Almeida and colleagues [39].

### 2.5. Bacterial Growth Inhibition Measurements

Bacterial growth inhibition was determined as the average diameter (in mm) of the inhibition zones around the spent media drops, measured at 3 different points of each drop [42].

### 2.6. Determination of Bacteriocin Concentration

Bacteriocin concentration was estimated in arbitrary units (AU) per mL for CFS and spent media at 0, 6, 12, 24, 48, and 72 h. The inhibitory dilutions were determined by plating 5 µL of serial dilutions onto lawns of sensitive strains and incubating overnight at 37 °C. Then, the inhibition zone was determined as described above. A bacteriocin relative concentration was obtained by the ratio of the length of the inhibition zone resulting from spent media and CFS samples [43].

### 2.7. Screening for Siderophore Production

Siderophore production by *E. coli* strains was assessed by using the chrome azurol S (CAS) solid medium assay, prepared as described by Schwyn and Neilands, 1987 [44].

### 2.8. PCR Amplifications

Genomic DNA from *E. coli* strains listed in Appendix A was prepared with gDNA Isolation kit (NZYtech, Lisboa, Portugal). Primers were designed based on colicin sequences available in NCBI (X87835.1, X82682.1, U15630.1, JN176450.1, HE603113.1, FJ664721.1, FJ664720.1, GU371926.1, FJ664734.1, FJ664743.1, J01574.1, HE603113.1, FJ004638.1, X02397.1, FJ985252.1, FJ573246.1, X63620.1, CP003111.1, X01009.1, CP001847.1, U15624.1, U15620.1, FJ664757.1, JX077110.1, FJ664729.1, and FJ664747.1). After the alignment of colicin sequences of activity genes using the bioinformatic tool Sequencher 5.0 (Gene Codes Corporation), they were divided into 6 groups (Ia, Ib; E1-E3, E6, E8-E9; E7; 5, 10; B and M) based on the similarity level (Appendix A) and primers for each group were designed. Amplification was performed in the Bio-Rad MyCycler under the following conditions: initial denaturation at 94 °C for 2 min followed by 30 cycles of 94 °C for 20 s, 59 °C for 20 s and 68 °C for 1 min, and the final elongation step at 72 °C for 7 min. The PCR products were examined by electrophoresis on 1% (*w*/*v*) agarose gels at 100 V and visualized by GelRed (Biotium, Fremont, CA, USA) staining on a Bio-Rad ChemiDoc XRS+ imager using the Image Lab software (version 4.0, BioRad, Hercules, CA, USA).

### 2.9. RNA Extraction

Biofilm biomass from *E. coli* CECT 504 and *E. coli* vs. *S. dysenteriae* single- and dual-species biofilm assays, was recovered at different time points (2, 6, and 24 h). At each time point, the biomass was scrapped using a sterile tip, the biofilm cells were collected, and then pelleted by centrifugation at 7000 rpm for 10 min at 4 °C. The total RNA was then extracted and purified using a PureLink™ RNA Mini Kit (Invitrogen, Waltham, MA, USA) and eluted in 50 μL of RNase-free water (Invitrogen, Waltham, MA, USA) following the manufacturer’s instructions. Afterward, the DNA contamination was removed by treating the samples with a DNase I (Invitrogen, Waltham, MA, USA) for 15 min, which was later inactivated by incubating for 10 min at 65 °C in the presence of 25 mM EDTA. The concentration and purity of the total RNA were spectrometrically assessed using a NanoDrop 1000™ (Thermo Scientific, Waltham, MA, USA). RNA integrity was evaluated by running a 1% (*w*/*v*) agarose gel in TAE electrophoresis buffer (40 mM Tris-HCl pH 7.2, 500 mM sodium acetate, and 50 mM EDTA), stained with SYBR^®^ Safe (Invitrogen, Waltham, MA, USA) and visualized by UV illumination (BioRad, Hercules, CA, USA).

### 2.10. Real-Time PCR (qPCR)

After RNA extractions, samples (50 ng/µL) from single- and dual-species biofilm experiments were reverse transcribed into cDNA at 42 °C for 30 min in a 20 µL reaction in the presence of iScript reaction mix and iScript reverse transcriptase, as described in the iScript™ cDNA Synthesis Kit (Bio-Rad). Afterward, quantification of the cDNA template (80 ng/µL) was performed using a qPCR reaction with a CFX96™ thermocycler (Bio-Rad), by using a Fast™ Evagreen Supermix 2x mix (Bio-Rad) and 0.5 µM of oligodeoxynucleotide primers. Primers were designed using the PRIMER3 software to specifically target colicin genes B, M, and E (provided in Appendix A). The cycling parameters were initially 98 °C for 2 min, followed by 40 repeats of 10 s at 98 °C, 10 s at 55 °C (see the annealing temperature of each set of primers in Appendix A), and finally, 5 s at 65 °C. The amplification of a standard housekeeping gene—GAPDH—was used to normalize the data [45,46]. As negative controls for each experiment, a nonreaction template (total RNA) and a negative control (RNase-free water) were included to ensure the absence of genomic DNA contamination. The number of colicin genes transcripted was expressed as the n-fold difference relative to the control gene, using the delta Ct method (2^ΔCt^), a variation of the Livak and Schmittgen method [47]. For each condition studied, three separate experiments with duplicates were performed.

### 2.11. Statistical Analysis

The data were analyzed using the statistical package GraphPad Prism version 6 (San Diego, CA, USA) by two-way ANOVA (Sidak’s multiple comparisons tests) since the data follow a normal distribution according to Kolmogorov–Smirnov’s test. Values with a *p* value < 0.05 were considered statistically significant.

## 3. Results

### 3.1. E. coli Biofilm Formation Ability, Bacteriocin Synthesis, and Siderophore Production

We started with the general characterization of all *E. coli* strains for properties that confer competitive advantages to bacteria. One of such characteristics is their ability to form biofilm, which is believed to be closely related to bacteria virulence [15,17]. Thus, *E. coli* strains listed in Appendix A (except *E. coli* Δ*impA* and *E. coli* Δ*entF*) were grown on 96-well tissue culture plates in LB medium and LB medium supplemented with 0.25% (*w*/*v*) glucose, for 24 h, and biofilms were quantified using the crystal violet method. An OD_570nm_ of 0.09 was chosen as the cut-off for biofilm-forming and non-forming strains, according to the criteria defined by Stepanović et al., 2000 [17]. Strains with OD between 0.09 < OD ≤ 0.18 were considered weak biofilm-formers and those with 0.18 < OD ≤ 0.36 were classified as being moderate biofilm-formers. *E. coli* CECT 434, CECT 504, and CECT 744 strains have demonstrated the major biofilm-forming capacity, followed by strains CECT 533, CECT 730, CECT 352, CECT 740, CECT 4783, and N5, which demonstrated a weak capacity to form a biofilm (Table 1). Eleven pathogenic *E. coli* strains were unable to form biofilm under these conditions (Appendix A). The addition of 0.25% (*w*/*v*) glucose did not potentiate the biofilm formation.

The majority of the strains, grown in 24 h planktonic culture, did not produce detectable levels of antimicrobial compounds against the tested organisms (Table 2). The strains considered good biofilm-formers were also tested for bacteriocin production in biofilm; however, the majority of strains with weak (+) and moderate (++) biofilm-forming capacity were not able to produce antimicrobial compounds in biofilm. The *E.*
*coli* CECT 504 was the only exception where bacteriocin synthesis was detected in 24-old biofilms. Since siderophores and colicins might compete for the same cell membrane receptors, the strain’s ability to produce siderophores was assessed by using CAS agar assay (Table 1). Of the 20 strains, 16 produced high levels of iron chelators.

### 3.2. Screening for Colicin Genes and Antimicrobial Spectrum

Primer sets for six groups of *E. coli* bacteriocins–colicins were designed based on sequences available in NCBI. PCR was performed with these primers and the presence of amplicons and their respective sizes were evaluated by gel electrophoresis (Figure 1). This allowed us to connect the growth inhibition effect with colicin synthesis. Based on screening results, the most frequent colicins were type II colicins [18,19]-B, M, Ia, and Ib (Table 1). The sequence of the amplicons obtained by PCR, using DNA of *E. coli* CECT 504 and primers for colicin B genes (Figure 1), was confirmed by sequencing (Stab Vida, Portugal). Type I colicins from group E-E1-3, E6, E8, and E9 were found only in CECT 504 and CECT 727 strains. It was not possible to amplify colicin E7-coding genes from any of the strains screened. As was expected, mutant *E. coli* strains and K12 strains that tested as sensitive in all soft agar experiments were not carrying colicin genes and, subsequently, genes for respective immunity proteins.

In order to investigate the spectrum of antimicrobial activity of *E. coli*, two biofilm-producing strains (CECT 504 and CECT 434) capable of inhibiting the growth of *E. coli* strain K12 (Table 2), but which antimicrobial molecules should be different (as only the CECT 504 tested positive for colicins), were tested for antimicrobial effect against the additional *E. coli* strains, and other Gram-negative and Gram-positive bacteria (see Appendix A). *E. coli* CECT 504 strain caused a growth inhibition effect of the largest number of *E. coli* strains tested (Table 2). Moreover, it was able to cause slight inhibition of *S. dysenteriae,* but no antimicrobial effect against other species was detected. On the other hand, *E. coli* CECT 434 only produced bacteriocins in the planktonic phase and its antimicrobial effect was less pronounced and limited to the *E. coli* species.

### 3.3. Bacteriocin Production during Biofilm Development

Aiming to investigate how bacteriocin production varies during biofilm development, we selected *E. coli* CECT 504. This strain displayed good biofilm-formation capacity, the ability to produce colicins both in planktonic and biofilm states and the capacity to inhibit the growth of the largest number of *E. coli* strains, also encoding the largest number of colicin genes (E1-3, E6, E8, E9, B, and M) comparing to other *E. coli* tested in the present study (Table 1 and Table 2). This strain was grown for 72 h and spent media samples were taken at different time points. In addition to the determination of biofilm biomass, we were also able to determine the relative concentration of the bacteriocin produced (AU/mL), using the assay described by Goebel et al. [43] (Figure 2). The production of bacteriocin has increased over 24 h, maintaining the levels of the bacteriocin similar after that in all serial dilutions.

No significant difference (*p* value > 0.05) was detected between biofilm growth with and without media changes, suggesting that media replacements have no influence on biofilm development and bacteria biomass formed (Figure 3). In addition, media replacement did not affect the amount and/or concentration of bacteriocin produced, since the AU observed for biofilm spent medium was similar for both conditions (data not shown). It was observed that the concentration of bacteriocin is similar for both conditions and increased drastically with the biofilm/culture age.

Interestingly, a decrease in the biofilm biomass was observed after 12 h of incubation in all experiments, which correlates with the higher amounts of bacteriocins. Since the colicin release normally requires cell lysis, and the lysis protein gene is located at the end of the colicin operon, the decrease in biofilm biomass can be evidence of active colicin production by *E. coli*.

### 3.4. Dual-Species Biofilms

To investigate the interactions between bacteriocin-producing *E. coli* CECT 504 and other closely related species, we formed dual-species biofilms with *S. dysenteriae* or *S. enterica*. As demonstrated in Figure 4, selective exclusion of *S. dysenteriae* by *E. coli* occurred in the dual-species biofilms.

While only a slight inhibitory effect of *E. coli* spent media on *S. dysenteriae* lawn (Table 2) was observed in the previous experiments, an antagonistic interaction between the two species was observed in biofilms. Despite no statistical difference (*p* value > 0.05) being found by comparing the values of *E. coli* and *S. dysenteriae* biofilm cells culturability (CFU/mL) with those for a single-species *E. coli* biofilm (Figure 4), it could be observed a decrease in the number of CFU/mL for *E. coli* grown in the dual-species biofilm that might be related to the increased production of colicin and thus increased rate of *E. coli* lysis.

Colicin-producing *E. coli* and *S. enterica* were able to coexist in dual-species biofilm, as can be observed in Figure 4. The proportion of CFUs/mL from these two species was approximately 35% to 65%, respectively.

### 3.5. Expression of Colicins in Dual-Species Biofilms

To confirm the hypothesis of *S. dysenteriae* exclusion due to bacteriocin synthesis and subsequent cell death, the expression of the three colicin genes previously identified on *E. coli* CECT 504 was evaluated (Figure 5). The housekeeping gene GAPDH was used to normalize the data. Since, in this strain, the residual expression of colicins was observed in single-species biofilms, gene expression values were compared between single- and dual-species biofilms. As such, an expression level of “1” does not mean an absence of expression; instead, it means that the residual expression of colicin genes is similar in single- and dual-species biofilms. While no significant changes were detected at an early stage (2 h) (*p* value > 0.05); colicin B, M, and E genes were significantly over-expressed for the 6 h of biofilm formation (*p* value < 0.05). For 24 h, as expected, the expression of colicin genes was restored to basal levels, which correlated with the absence of *Shigella* observed by the culturability data (Figure 4). This indicates that, as soon as a noncompetitive environment is present, the expression of colicin genes is restored to normal levels.

## 4. Discussion

In the present study, we characterized different strains of *E. coli,* a relevant foodborne pathogen, by assessing several properties that confer important competitive advantages against neighboring microorganisms. Two of such advantages are the biofilm formation capacity [48,49] and the secretion of antimicrobial molecules [23], which seems to be a relatively widespread phenomenon among bacteria and even among other microorganisms.

Interestingly, it has previously been reported that 35% of *E. coli* strains appearing in the human intestinal tract are colicinogenic [50], meaning that they are able to produce colicins. Thus, in order to screen for the presence of genes responsible for the biosynthesis of colicins, and since no full-genome sequencing data were available for these *E. coli* strains, we resorted to PCR. Colicin genes were found to be widely distributed among pathogenic *E. coli* strains (Table 1). Their production on planktonic phase or 24 h single-species biofilms was uncommon. This might very well be due to the fact that the colicin biosynthetic operon is under SOS promoter regulation; thus, colicin production seems to be induced during stressful conditions.

Additionally, colicins Ia, Ib, B, M, 5, and 10, which belong to the group of nonsecreted colicins [25,50], were found in the majority of the tested strains. As nonsecreted molecules, their antimicrobial effect might not be easily observed on the supernatant, since only residual amounts of colicins might be released as a consequence of cell death in the stationary phase. In opposition, we also observed some antimicrobial effects on supernatants from two *E. coli* strains not coding for colicin genes—strains CECT 434 and CECT 730. This effect could be due to the production of microcin, another type of *E. coli* bacteriocin, which is not SOS-dependent [31].

Our results also suggest that resistance genes of some specific colicins might confer resistance to other, probably closely related, colicins groups, as they share the mode of action. For example, only genes for colicins Ia and Ib were detected in *E. coli* CECT 515T but it was resistant to the effect of colicins produced by the CECT 504 strain. Besides genes for colicins B and M, genes for colicins 5 and 10 were detected in *E. coli* CECT 4555 strain, which is also resistant to CECT 504 colicins action. This might be possible due to the fact that colicins Ia, Ib, 5, and 10 are pore-forming as colicin B and E1 (produced by CECT 504 strain) are, thus, sharing the same mode of action [25].

Regarding siderophores, almost all *E. coli* strains tested produced siderophores. At the same time, the majority of these were found to be resistant to the antibacterial molecules (Table 1). It has been proposed that this might happen as the uptake of colicins in Gram-negative bacteria may occur through siderophore receptors [25,51,52]. This means that some *E. coli* strains, able to synthesize their own siderophores, are therefore not susceptible to the colicins effect due to the competition between the siderophores and the colicins for the same receptor (*tol* machinery for type I colicins and TonB-dependent transporters for type II colicins) [25,26,53,54]. On the other hand, the receptors of nonproducing strains are less busy with the uptake of siderophores, as they rely on the production of iron-chelator by other strains. Since here, all strains were growing separately, the receptors were completely available for the uptake of colicins molecules.

Concerning the colicins production by *E. coli* CECT 504 in a biofilm state, media replacements had no effect on the biofilm biomass formed and on the concentration of bacteriocin; but the decrease in bacterial biomass after 12 h of incubation indicates active colicin production. Its release normally requires *E. coli* cell lysis [25], also indicating that the inhibitory effect observed (Figure 2) was mainly due to type I colicins, in the case of this strain colicin E, and not to the presence of type II colicins or microcins.

When growing the colicin-producer *E. coli* CECT 504 together with *S. dysenteriae*, not found to produce any type of detectable antibacterials, the latter strain was selectively excluded from the biofilm. This result was supported by the detection of high levels of gene expressions responsible for colicin synthesis (B, M, and E). This exclusion phenomenon is, in fact, surprising since previous works have shown that bacteriocin-producing and nonproducing strains can coexist in the biofilms by forming individual microcolonies [55]. Aoki and colleagues [56] also demonstrated that, beyond growth inhibition by antimicrobial compounds, some pathogenic *E. coli* are able to establish contact with other bacteria, consequently inhibiting their proliferation. This contact-dependent inhibition does not depend on antibacterial compound secretion. On the other hand, the slight decrease in *E. coli* cells in dual-species biofilm compared to single-species biofilm and the proven expression of colicin genes in old biofilms suggests colicin production, during which the cell lysis occurs. In fact, we might have a combined effect of both strategies.

A different situation was observed when *E. coli* CECT 504 was grown in the dual-species biofilm with *S. enterica*, with the two *Enterobacteriaceae* species being able to coexist probably as a result of *Salmonella* resistance to *E. coli* bacteriocins. Cells in the biofilm are surrounded by a self-produced matrix creating heterogeneous structures in which chemical gradients in the concentrations of nutrients and metabolic products determine individual niches of microbes [57,58,59]. This way, different species can coexist in biofilms [60,61] by reaching a new balance within the dual-species population. For these particular species combinations, two opposite behaviors were observed. Competing species were either completely excluded or allowed to form biofilm at concentrations levels similar to the single-species biofilm.

## 5. Conclusions

In conclusion, we observed that colicin genes are widely distributed in the *E. coli* strains tested in the present study, and most likely beyond these. Despite no tight correlation being found between the production of colicins and/or siderophores, and the biofilm-forming ability, it was clear that the largest amounts of bacteriocin were produced in the late stages of biofilm development. This was in agreement with the observation that a bacteriocin-producer strain selectively excluded closely related species from dual-species biofilm, and correlates well with the gene expression data obtained for dual-species biofilms. With this study, it also becomes clear the need to better understand the dynamics of interspecies relations within biofilm communities and their signaling, which is what we believe this work contributed to.

## Figures and Tables

**Figure 1 foods-11-02652-f001:**
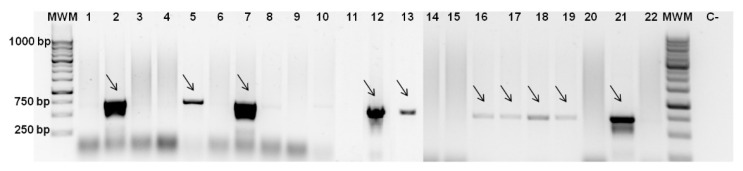
PCR for colicin genes detection. PCR was performed using primers for colicin B genes and gDNA from *E. coli* strains: 1- CECT 352, 2- CECT 504, 3- CECT 515, 4- CECT 730, 5- CECT 740, 6- CECT 744, 7- CECT 832, 8- CECT 4267, 9- K12 *ΔimpA*, 10-ER1100A Δ*entF*, 11- CECT 533, 12- CECT 727, 13- CECT 4782, 14- CECT 434, 15- CECT 736, 16- CECT 4537, 17- CECT 4555, 18- CECT 4783, 19- 5947, 20- K12, 21- N5, 22- NCTC 12900. C- negative control, MWM–molecular weight marker. Agarose gel (1% *w/v*) electrophoresis was carried out at 100 V. All PCR products had the appropriate size of 709 bp and were assigned with arrows.

**Figure 2 foods-11-02652-f002:**
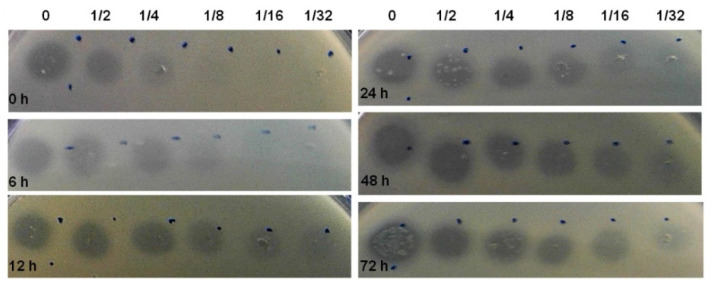
Growth inhibition zones. Soft agar test performed on lawns of *E. coli* K12. Free-cell supernatants of *E. coli* CECT 504 biofilms, taken at 0, 6, 12, 24, 48, and 72 h, were serially diluted (to 1/32), 5 µL were applied on lawns, and the plates were incubated at 37 °C overnight.

**Figure 3 foods-11-02652-f003:**
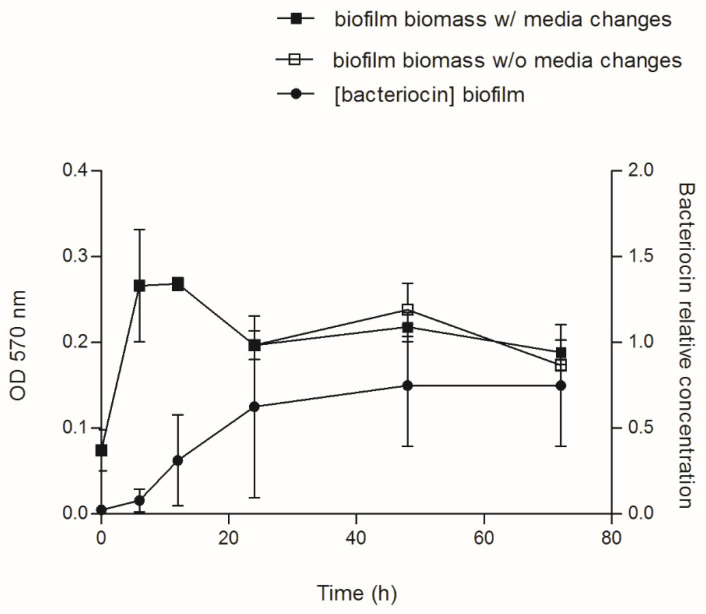
Biofilm biomass and bacteriocin production during biofilm development. Single-species *E. coli* CECT 504 biofilms were grown in 24-well plates up to 72 h, with (beginning at 24 h) and without media changes. At 0, 6, 12, 24, 48, and 72 h, cell-free spent media from biofilms were tested for the antimicrobial effect, and the concentration of bacteriocin in arbitrary units per mL (AU/mL) was estimated.

**Figure 4 foods-11-02652-f004:**
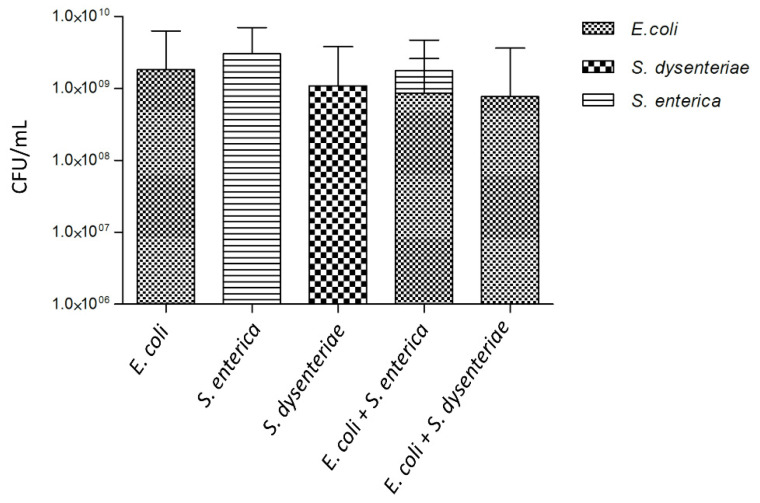
Biofilm formation for single- and dual-species biofilms. Single-species biofilms of *E. coli* CECT 504, *S. enterica* SGSC 2476, *S. dysenteriae* ATCC 11335, and dual-species biofilms of *E. coli* CECT 504 + *S. enterica* SGSC 2476 and *E. coli* CECT 504 + *S. dysenteriae* ATCC 11335 were grown in 24-well plate to 24 h and CFUs were counted on MacConkey agar plates.

**Figure 5 foods-11-02652-f005:**
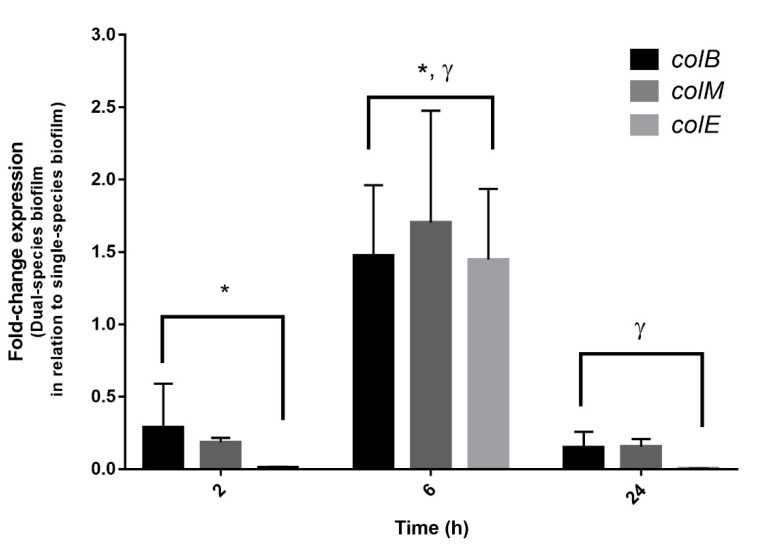
Expression of the genes involved in the colicin (B, M, and E) synthesis in dual-species biofilms of *E. coli* CECT 504 and *S. dysenteriae* ATCC 11335, at different time points. The amount of colicin genes transcribed is expressed as the n-fold of the relative between dual- and single-species biofilm experiments. Data were normalized with a standard housekeeping gene (GAPDH) under the same conditions. * Significant differences between 2 h and 6 h of incubation (*p* value < 0.05). ^γ^ Significant differences between 6 h and 24 h of incubation (*p* value < 0.05).

**Table 1 foods-11-02652-t001:** Characterization of *E. coli* strains.

	Biofilm Forming Capacity ^a^	Bacteriocin Production		Colicin Genes	Siderophore Synthesis ^c^
Strain		Planktonic Culture ^b^	Biofilm ^b^		
NCTC 12900	-	-	ND	Ia; Ib	+
CECT 4267	-	-	ND	Ia; Ib	+
CECT 4782	-	-	ND	B; M	+
CECT 4783	+	-	-	Ia; Ib; B; M	+
CECT 5947	-	-	ND	Ia; Ib; B; M	+
K12	-	-	ND	-	-
CECT 352	+	-	-	-	+
CECT 434	++	+	-	-	+
CECT 504	++	+	+	E1-3; E6; E8; E9; B; M	-
CECT 515T	-	-	ND	Ia; Ib	+
CECT 533	+	-	-	-	+
CECT 727	-	+	-	E1-3; E6; E8; E9; B; M	+
CECT 730	+	+	-	-	+
CECT 736	-	-	ND	-	-
CECT 740	+	-	-	Ia; Ib; B; M	+
CECT 744	++	-	-	-	+
CECT 832	-	+	ND	Ia; Ib; B; M	-
CECT 4537	-	-	ND	Ia; Ib; B; M	+
CECT 4555	-	+	ND	5; 10; B; M	+
N5	+	+	-	Ia; Ib; 5; 10; B; M	+

^a^ No biofilm-forming capacity (-), weak to moderate biofilm-forming capacity (+/++). ^b^ No bacteriocin production and absence of inhibition zone (-), bacteriocin production and presence of inhibition zone (+). ND–Not determined (only strains able to form biofilms were tested for bacteriocin production in biofilm). ^c^ No siderophore production (-), siderophore production (+).

**Table 2 foods-11-02652-t002:** Growth inhibition effect of biofilm-forming *E. coli* (CECT 504 and CECT 434) strains.

		Growth Inhibition Zone Diameter (mm)					
		Bacteria Lawns					
	Spent Media	*E. coli* K12	*E. coli* CECT 352	*E. coli* CECT 736	*E. coli* CECT 744	*E. coli* NCTC 12900	*S. dysenteriae* ATCC 11335
24 h-biofilm	CECT 504	10.40 ± 2.26	6.00 ± 0.00	5.33 ± 0.47	7.00 ± 0.00	6.00 ± 0.00	*
	CECT 434	-	-	-	-	-	-
24 h-planktonic culture	CECT 504	9.30 ± 1.06	6.33 ± 0.29	-	-	-	*
	CECT 434	-	4.00 ± 0.00	4.83 ± 0.29	4.17 ± 0.29	4.83 ± 0.29	-

* Slightly distinct inhibition zone; it was not possible to measure inhibition zone diameter. – Absence of inhibition zone. “±” means standard deviation.

## Data Availability

The data presented in this study are available within the article and Appendix A.

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
