# Peer review of "Bacteriocin Production by Escherichia coli during Biofilm Development"

_foods, 2022, doi:10.3390/foods11172652_

Round 1
Reviewer 1 Report
Dear editors and authors
The manuscript (Bacteriocin production by Escherichia coli during biofilm development) needs many corrects and modifications.
1- Some linguistic terms do not fit with scientific writing such as Page 1 line 21 (neighboring) , Page 4 line 125 (consume), ....... etc. , so there should be a comprehensive review of the manuscript language.
2- In order to support the manuscript introduction, the authors are suggested to add two references (1- Al-kuzayi, A. K., & Al-Sahlany, S. T. (2011). DETECTING FOR E. COLI O157: H7 IN DAIRY PRODUCTS WHICH WERE LOCALLY PROCESSED AND FOUND IN BASRA CITY MARKETS. Basrah Journal of Agricultural Sciences, 24(1):289-303.
2-Verma, D. K., Thakur, M., Singh, S., Tripathy, S., Gupta, A. K., Baranwal, D., ... & Srivastav, P. P. (2022). Bacteriocins as antimicrobial and preservative agents in food: Biosynthesis, separation and application. Food Bioscience, 46, 101594.)
3- Many working methods do not have references such as Antimicrobial activity assay, Cultur ability assessment of dual-species biofilms, and Bacterial growth inhibition measurements.
4-Page 2 line 74-75, Did you use strains or isolates? What are the sources of these bacterial strains? How many of these strains should this be written in the working methods? Here the ways work is not clear. What is the number of plasmid used? Where did these plasmids come from?
5-Page 2 line 76, Reference 29 Work on Bacillus thuringiensis, What is the relationship between the genes of these bacteria (E. coli)?
6-Page 2 line 80 Add new reference of bacteria growth method, I suggest you (Sezonov, G., Joseleau-Petit, D., & d'Ari, R. (2007). Escherichia coli physiology in Luria-Bertani broth. Journal of bacteriology, 189(23).
7-Page 3 line 84 and line 91, The unit of microorganism is CFU/mL, it is not Cell / mL Please correct.
8-Page 3 line 112, How many numbers of bacteria in 50 μL??
9-Some method work is unclear such as Determination of bacteriocin concentration.
10-Page 4 line 134, What is this ? 5 μL or drops!!!!!
11-In All manuscript , Please correct the halo to inhibition zone.
12-Table 2, Add + standard deviation below the table.
13-Figure 3 The y-axis, why use this wavelength? Nothing was mentioned in the working methods!!!!
Author Response
We would like to express our appreciation to the reviewer for their careful reading of the text and for all the in-depth constructive comments and suggestions about the first version of our manuscript entitled “Bacteriocin production by Escherichia coli during biofilm development” with ref. foods-1863955. We have put an earnest effort to respond to the concerns of each reviewer in detail and we are now submitting a revised manuscript.
We are confident that the changes performed are in accordance with the reviewer requests and hope the reviewer feels that our manuscript is now acceptable for publication.
A point-by-point description of our answers to the reviewer’ comments and suggestions follows. The revised manuscript has all changes highlighted in yellow to allow better follow-up by the reviewer.
Comments from the reviewers
Reviewer 1
Q1. “Some linguistic terms do not fit with scientific writing such as Page 1 line 21 (neighboring), Page 4 line 125 (consume), ....... etc., so there should be a comprehensive review of the manuscript language.”
Authors answer#1. We apologize for the use of some linguistic terms that do not fit with scientific writing. As such, we have revised this point throughout the manuscript.
Q2. “In order to support the manuscript introduction, the authors are suggested to add two references (1- Al-kuzayi, A. K., & Al-Sahlany, S. T. (2011). DETECTING FOR E. COLI O157: H7 IN DAIRY PRODUCTS WHICH WERE LOCALLY PROCESSED AND FOUND IN BASRA CITY MARKETS. Basrah Journal of Agricultural Sciences, 24(1):289-303.
2-Verma, D. K., Thakur, M., Singh, S., Tripathy, S., Gupta, A. K., Baranwal, D., ... & Srivastav, P. P. (2022). Bacteriocins as antimicrobial and preservative agents in food: Biosynthesis, separation and application. Food Bioscience, 46, 101594.)”
Authors answer#2. We appreciate the reviewer suggestion, and, as such, we have added these 2 new references. Additionally, 5 more references (from the last 5 years) were also added in the introduction section of the revised manuscript.
References used to support this comment:
- Eick, S. Biofilms. Monographs in Oral Science 2021, 29, 1–11, doi:10.1159/000510184.
- Castro, J.; França, A.; Bradwell, K.R.; Serrano, M.G.; Jefferson, K.K.; Cerca, N. Comparative transcriptomic analysis of Gardnerella vaginalis biofilms vs. planktonic cultures using RNA-Seq. npj Biofilms and Microbiomes 2017, 3, doi:10.1038/s41522-017-0012-7.
- Jin, X.; Kightlinger, W.; Kwon, Y.C.; Hong, S.H. Rapid Production and Characterization of Antimicrobial Colicins Using Escherichia coli-Based Cell-Free Protein Synthesis. Synthetic Biology 2018, 3, doi:10.1093/synbio/ysy004.
- Meade, E.; Slattery, M.A.; Garvey, M. Bacteriocins, Potent Antimicrobial Peptides and the Fight against Multi Drug Resistant Species: Resistance Is Futile? Antibiotics 2020, 9. doi:3390/antibiotics9010032
- Niamah, A.K.; Thyab, S.; Al-Sahlany, G. Detecting for coli O157:H7 in dairy products which were locally processed and found in Basra city markets. Basrah Journal of Agricultural Sciences 2011, 24, 290-299.
- Verma, D.K.; Thakur, M.; Singh, S.; Tripathy, S.; Gupta, A.K.; Baranwal, D.; Patel, A.R.; Shah, N.; Utama, G.L.; Niamah, A.K.; et al. Bacteriocins as Antimicrobial and Preservative Agents in Food: Biosynthesis, Separation and Application. Food Bioscience 2022, 46, 101594, doi:10.1016/J.FBIO.2022.101594.
- WasiÅ„ski, B. Extra-Intestinal Pathogenic Escherichia Coli – Threat Connected with Food-Borne Infections. Annals of Agricultural and Environmental Medicine 2019, 26, 532–537. doi:10.26444/aaem/111724.
Q3. Many working methods do not have references such as Antimicrobial activity assay, Cultur ability assessment of dual-species biofilms, and Bacterial growth inhibition measurements.
Authors answer#3. We appreciate the reviewer suggestion, and, as such, we have now added appropriate references to each method as pointed out by the reviewer in this question.
References used to support this comment:
- Almeida, C.; Azevedo, N.F.; Santos, S.; Keevil, C.W.; Vieira, M.J. Discriminating Multi-Species Populations in Biofilms with Peptide Nucleic Acid Fluorescence in Situ Hybridization (PNA FISH). PLoS ONE 2011, 6, e14786. doi:10.1371/journal.pone.0014786.
- Alsop, G.M.; Waggy, G.T.; Conway, R.A. Bacterial Growth Inhibition Test. Journal (Water Pollution Control Federation) 1980, 25, 2452-2456.
Q4. Page 2 line 74-75, Did you use strains or isolates? What are the sources of these bacterial strains? How many of these strains should this be written in the working methods? Here the ways work is not clear. What is the number of plasmid used? Where did these plasmids come from?
Authors answer#4. We apologize if we were unclear in the submitted version of the manuscript. As such, we have introduced the source of the strains/isolate, as can be seen in Supplementary Material – Table S1. In fact, the used strains from a wide culture collection, namely: American Type Culture Collection; Colección Española de Cultivos Tipo; National Collection of Type Culture; and South Georgia State College. In addition, in this study, we have only used one isolate (E. coli N5), as shown in Table S1. Regarding the plasmids, there belong to our laboratory collection, as also mentioned in Table S1.
Q5. “Page 2 line 76, Reference 29 Work on Bacillus thuringiensis, What is the relationship between the genes of these bacteria (E. coli)?”
Authors answer#5. We thank the reviewer for noticing this lack of clarity. In fact, there was a mistake in the reference. As such, we have replaced the reference in the revised version of the manuscript.
Q6. “Page 2 line 80 Add new reference of bacteria growth method, I suggest you (Sezonov, G., Joseleau-Petit, D., & d'Ari, R. (2007). Escherichia coli physiology in Luria-Bertani broth. Journal of bacteriology, 189(23).”
Authors answer#6. We thank the reviewer for the suggestion, the reference was added to the sentence in the revised version of the manuscript.
Q7. “Page 3 line 84 and line 91, The unit of microorganism is CFU/mL, it is not Cell / mL Please correct.”
Authors answer#7. We have changed the manuscript according to the reviewer correction.
Q8. “Page 3 line 112, How many numbers of bacteria in 50 μL??”
Authors answer#8. The reviewer has a valid point here. We used a concentration of 5.0 ×105 CFU/mL in 50 μL of bacterial suspension, and this information was added in the revised version of the manuscript.
Q9. “Some method work is unclear such as Determination of bacteriocin concentration.”
Authors answer#9. We apologize for the lack of clarity. In order to be more elucidate, we have changed the revised version of the manuscript, as follows: “Bacteriocin concentration was estimated in arbitrary units (AU) per mL for Cell Free Supernatant (CFS) and spent media at 0, 6, 12, 24, 48 and 72 h. The inhibitory dilutions were determined by plating 5 µL of serial dilutions onto lawns of sensitive E. coli CECT504 and incubating overnight at 37 °C. Then, the inhibition zone was determined as described above. A bactericion relative concentration was obtained by the ratio of the length of inhibition zone resulting from spent media and CFS samples [34].”
Q10. “Page 4 line 134, What is this? 5 μL or drops!!!!!”
Authors answer#10. We apologize if we were unclear. In fact, we have used 5 μL, as such, we have changed the manuscript accordingly.
Q11. “In All manuscript, please correct the halo to inhibition zone.”
Authors answer#11. We thank the reviewer for the suggestion. We have now replaced “halo” by “inhibition zone”.
Q12. “Table 2, add + standard deviation below the table.”
Authors answer#12. Thank you for your observation. As such, we have added “±” as a note in Table 2.
Q13. “Figure 3 The y-axis, why use this wavelength? Nothing was mentioned in the working methods!!!!”
Authors answer#13. The reviewer has a valid point here. We have introduced a sentence in the methods section explaining that total biofilm biomass was determined at an OD of 570 nm. Indeed, this wavelength has been used to quantify total biofilm biomass as described in Almeida et al, 2011.
References used to support this comment:
- Almeida, C.; Azevedo, N.F.; Santos, S.; Keevil, C.W.; Vieira, M.J. Discriminating multi-species populations in biofilms with peptide nucleic acid fluorescence in situ hybridization (PNA FISH). PLoS ONE 2011, 6, e14786. https://doi.org/10.1371/journal.pone.0014786.
Reviewer 2 Report
The manuscript "Bacteriocin production by Escherichia coli during biofilm development" has an interesting result. However, the manuscript needs to be improved. Authors are suggested to check typos throughout the manuscript.
- L49: Extracellular DNA (eDNA)
- L44: Authors are suggested to use the most updated references throughout the introduction section (check publications from the last 5 years).
- L79: insert space between "37" and " ºC" and make corrections throughout the manuscript.
- L92-93: What was the reason to agitate the plates at 120 rpm during biofilm formation? Validate your answers with appropriate references.
- L105: 6500 × g
- L122: Sources (manufacture name, city, country) need to mention all the chemicals, reagents, and equipment used in this manuscript.
- “The last inhibitory dilutions” meaning not clear. Rephrase the sentence.
- L145: …. FJ664729.1, and FJ664747.1)
- L157: (2, 6, and 24 h).
- L150: Use the appropriate symbol for temperature (ºC).
- Many spacing problems are found in the tables.
- L315: no statistical difference (P value > 0.05).
Author Response
We would like to express our appreciation to the reviewer for their careful reading of the text and for all the in-depth constructive comments and suggestions about the first version of our manuscript entitled “Bacteriocin production by Escherichia coli during biofilm development” with ref. foods-1863955. We have put an earnest effort to respond to the concerns of each reviewer in detail and we are now submitting a revised manuscript.
We are confident that the changes performed are in accordance with the reviewer requests and hope the reviewer feels that our manuscript is now acceptable for publication.
A point-by-point description of our answers to the reviewer’ comments and suggestions follows. The revised manuscript has all changes highlighted in yellow to allow better follow-up by the reviewer.
Reviewer 2
“The manuscript "Bacteriocin production by Escherichia coli during biofilm development" has an interesting result. However, the manuscript needs to be improved. Authors are suggested to check typos throughout the manuscript.”
We appreciate the reviewer feedback, and we acknowledge the reviewer has carefully analyzed the manuscript.
Q1. “L49: Extracellular DNA (eDNA)”
Authors answer#1. Thank you for your observation. As such, the description of eDNA was added to the sentence in the revised version of the manuscript.
Q2. “L44: Authors are suggested to use the most updated references throughout the introduction section (check publications from the last 5 years)”.
Authors answer#2. We appreciate the reviewer suggestion, and, as such, we have added 5 new references from the last 5 years, in the introduction section of the revision of the manuscript.
References used to support this comment:
- Eick, S. Biofilms. Monographs in Oral Science 2021, 29, 1–11, doi:10.1159/000510184.
- Castro, J.; França, A.; Bradwell, K.R.; Serrano, M.G.; Jefferson, K.K.; Cerca, N. Comparative transcriptomic analysis of Gardnerella vaginalis biofilms vs. planktonic cultures using RNA-Seq. npj Biofilms and Microbiomes 2017, 3, doi:10.1038/s41522-017-0012-7.
- Jin, X.; Kightlinger, W.; Kwon, Y.C.; Hong, S.H. Rapid Production and Characterization of Antimicrobial Colicins Using Escherichia coli-Based Cell-Free Protein Synthesis. Synthetic Biology 2018, 3, doi:10.1093/synbio/ysy004.
- Meade, E.; Slattery, M.A.; Garvey, M. Bacteriocins, Potent Antimicrobial Peptides and the Fight against Multi Drug Resistant Species: Resistance Is Futile? Antibiotics 2020, 9. doi:3390/antibiotics9010032
- Verma, D.K.; Thakur, M.; Singh, S.; Tripathy, S.; Gupta, A.K.; Baranwal, D.; Patel, A.R.; Shah, N.; Utama, G.L.; Niamah, A.K.; et al. Bacteriocins as Antimicrobial and Preservative Agents in Food: Biosynthesis, Separation and Application. Food Bioscience 2022, 46, 101594, doi:10.1016/J.FBIO.2022.101594.
- WasiÅ„ski, B. Extra-Intestinal Pathogenic Escherichia coli – Threat Connected with Food-Borne Infections. Annals of Agricultural and Environmental Medicine 2019, 26, 532–537. doi:10.26444/aaem/111724.
Q3. “L79: insert space between "37" and " ºC" and make corrections throughout the manuscript.”
Authors answer#3. We thank the reviewer for noticing this mistake. We have now revised the manuscript to correct the mentioned grammar mistake.
Q4. “L92-93: What was the reason to agitate the plates at 120 rpm during biofilm formation? Validate your answers with appropriate references.”
Authors answer#4. The reviewer has a valid point here. As well-established, agitation is essential to promote an optimal attachment of the cells, the first step of biofilm formation, and to enhance oxygenation (Paweli et al, 2018). The agitation of 120 rpm is commonly used to form E. coli biofilm as described by Almeida and colleagues (2011).
References used to support this comment:
- Almeida, C.; Azevedo, N.F.; Santos, S.; Keevil, C.W.; Vieira, M.J. Discriminating multi-species populations in biofilms with peptide nucleic acid fluorescence in situ hybridization (PNA FISH). PLoS ONE 2011, 6, e14786. https://doi.org/10.1371/journal.pone.0014786.
- Paweli, N.E..; Suryadarma, P.; Mardhiati, I.; Mangunwidjaja, D. The role of oxygenation on the attachment of Escherichia coli rpoS mutant under aerobic conditions. AIP Conference Proceedings 2018, 2049, 020015, org/10.1063/1.5082420.
Q5. “L105: 6500 × g”
Authors answer#5. We thank the reviewer for noticing this mistake. We have now revised the manuscript to correct the mentioned grammar mistake.
Q6. “L122: Sources (manufacture name, city, country) need to mention all the chemicals, reagents, and equipment used in this manuscript.”
Authors answer#6. We apologize for missing the source of the chemicals. As such, we have added this information to the revised version of the manuscript.
Q7. “The last inhibitory dilutions” meaning not clear. Rephrase the sentence.
Authors answer#7. We apologize for the lack of clarity in the description of the method: determination of bacteriocin concentration. In order to be more elucidate, we have changed the revised version of the manuscript, as follows: “Bacteriocin concentration was estimated in arbitrary units (AU) per mL for Cell Free Supernatant and spent media at 0, 6, 12, 24, 48 and 72 h. The inhibitory dilutions were determined by plating 5 µL of serial dilutions onto lawns of sensitive E. coli CECT504 and incubating overnight at 37 °C. Then, the inhibition zone was determined as described above. A bacteriocin relative concentration was obtained by the ratio of the length of inhibition zone resulting from spent media and CFS samples [34].”
Q8. “L145: …. FJ664729.1, and FJ664747.1)”
Authors answer#8. We thank the reviewer for noticing this mistake. We have now revised the manuscript to correct the mentioned grammar mistake.
Q9. “L157: (2, 6, and 24 h).”
Authors answer#9. We thank the reviewer for noticing this mistake. We have now revised the manuscript to correct the mentioned grammar mistake.
Q10. “L150: Use the appropriate symbol for temperature (ºC).”
Authors answer#10. We thank the reviewer for noticing this mistake. We have now revised the manuscript to correct the mentioned grammar mistake.
Q11. “Many spacing problems are found in the tables.”
Authors answer#11. The tables are now formatted, and the spaces were removed in the revised version of the manuscript.
Q12. “L315: no statistical difference (P value > 0.05).”
Authors answer#12. We thank the reviewer for noticing this mistake. We have now revised the manuscript to correct the mentioned grammar mistake.
Round 2
Reviewer 1 Report
Dear Editors,
The authors made all necessary modifications to improve the work, and I now suggest that it be published in its current form.